# Motivation for Vaccination against COVID-19 in Persons Aged between 18 and 60 Years at a Population-Based Vaccination Site in Manresa (Spain)

**DOI:** 10.3390/vaccines10040597

**Published:** 2022-04-12

**Authors:** Glòria Sauch Valmaña, Aïna Fuster-Casanovas, Anna Ramírez-Morros, Berta Rodoreda Pallàs, Josep Vidal-Alaball, Anna Ruiz-Comellas, Queralt Miró Catalina

**Affiliations:** 1Unitat de Suport a la Recerca de la Catalunya Central, Fundació Institut Universitari per a la Recerca a l’Atenció Primària de Salut Jordi Gol i Gurina (IDIAPJGol), 08272 Barcelona, Spain; afuster.cc.ics@gencat.cat (A.F.-C.); amramirez.cc.ics@gencat.cat (A.R.-M.); brodoreda.cc.ics@gencat.cat (B.R.P.); jvidal.cc.ics@gencat.cat (J.V.-A.); aruiz.cc.ics@gencat.cat (A.R.-C.); qmiro.cc.ics@gencat.cat (Q.M.C.); 2Health Promotion in Rural Areas Research Group, Gerència Territorial de la Catalunya Central, Institut Català de la Salut, 08272 Barcelona, Spain; 3Facultat de Medicina, Universitat de Vic Universitat Catalunya Central, 08500 Vic, Spain

**Keywords:** vaccination, COVID-19, motivation, health beliefs, pandemic

## Abstract

Our purpose was to identify the reasons why members of the population, aged 18–60 years, are vaccinated against COVID-19 at the mass vaccination point in Bages, Spain. This is 1 of 42 provisional spaces outside of health centres which have been set up in Catalonia in the context of the COVID-19 pandemic, and where people from all over Catalonia could go to be vaccinated by appointment. Methodology: We performed a cross-sectional study of users attending mass vaccination points in Bages during the months of July, August, and September 2021. Results: A total of 1361 questionnaires were statistically analysed. The most common reasons for vaccination were fear of infecting family (49.52%) and fear of self-infection (39.45%), followed by socialising (31.00%) and travel (30.56%). However, by applying a logistic regression model to each reason for vaccination, it was possible to estimate the associations regarding age, sex, marital status, educational level, production sector, mass vaccination point, previous COVID-19 infection, and COVID-19 infection of a family member. Relevance: The data generated will inform decisions and formulations of appropriate campaigns that will promote vaccination in specific population groups.

## 1. Introduction

The 2019 coronavirus disease (COVID-19), caused by severe acute respiratory syndrome coronavirus-2 (SARS-CoV-2), has had a major impact on the global health and economy since its emergence in late 2019. Vaccination against SARS-CoV-2 is the main strategy against COVID-19 worldwide [1]. It protects against serious pathology, hospitalisation, and death, and it reduces the risk of human-to-human transmission. Protection against new variants of the virus may be lower, but protection against severe disease and death remains high. In order to achieve high vaccination coverage, it is necessary to encourage and raise awareness among the population to be vaccinated. Vaccination lies at the intersection between the individual and society and involves a balance between an individual’s decision to accept or refuse a vaccine and the public health benefits derived from herd immunity when large numbers of people are vaccinated. For optimal success, vaccination programmes need a high level of uptake [2].

Vaccine hesitancy is complex; the literature highlights that it is driven by both individual factors (emotions, values, risk perceptions, knowledge, or beliefs) and social, cultural, political, and historical factors [3,4]. The pandemic has increased awareness of the importance of vaccines for the vast majority of vaccine-accepting people. Research has shown that newer vaccines generate greater hesitancy [5].

Results from cross-sectional surveys among representative samples of adults in high-income countries indicate that the vast majority (67–73.4%) are willing to be vaccinated against COVID-19 [6,7]. A European study conducted in Denmark, France, Germany, Italy, Portugal, the Netherlands, and the United Kingdom concluded that willingness to be vaccinated is higher among men over 55 years of age. Men who are unwilling to be vaccinated are younger, with a higher proportion between the ages of 18 and 24 years. Uncertainty among women is higher in all age groups and higher in women aged 45–54 years [7]. A U.S. study, conducted through surveys every 14–28 days on an internet platform, concluded that vaccine hesitancy has undergone changes during the course of the pandemic [8]. After an increase in vaccine reluctance in 2020 [9,10], this study showed a longitudinal decline in vaccine hesitancy in late 2020 and early 2021 across all demographic groups, especially among Black and Hispanic participants, who had experienced a disproportionate burden of serious illness and death from COVID-19 [11,12]. The reduced hesitancy occurred in conjunction with the regulatory approval of COVID-19 vaccines and the roll-out of mass vaccination programmes. Despite these gains, in March 2021, estimates of vaccine hesitancy remained elevated, especially among young adults, Black participants, and those of low socio-economic status [8].

These studies show that considerable political effort may be required to achieve adequate vaccination rates. At the beginning of July, 40.80% of people in Catalonia were vaccinated with a complete schedule, while 56.13% were vaccinated with the first dose. During the summer, efforts were intensified to accelerate vaccination, and by the end of September, coinciding with the closing of the mass vaccination points, 76.16% of people were vaccinated with a complete schedule, and 73.69% were vaccinated with the first dose.

It is important to determine the health motivations that contribute to the decision to be vaccinated. By knowing the health beliefs that promote the acceptance of vaccination, appropriate campaigns can be formulated to promote vaccination in specific population groups.

During the pandemic caused by COVID-19, mass vaccination centres have been opened, sites normally used for non-health activities, which have allowed for the vaccination of a large volume of people at a high speed [13]. The main objective of this study is to know the main motivations for vaccination against COVID-19 of the adults under age 60 at the mass vaccination point in Manresa during the months of July–September 2021.

## 2. Materials and Methods

### 2.1. Study Design

A descriptive, cross-sectional design was employed, involving a voluntary participation survey addressed to users between 18 and 60 years of age who came to be vaccinated against COVID-19 (regardless of the commercial brand of the vaccine administered) at the mass vaccination point in Manresa (Spain) from 1 July to 30 September 2021.

### 2.2. Sample

A sample of 1333 individuals was estimated to be necessary to estimate the reasons for vaccination with a precision of 3 percentage points and 95% confidence intervals. It was estimated that an excess of 20% of the population would be necessary in case replacements were needed. The authors used Grammo software (version 7.12, IMIM, Barcelona, Spain). Because of a lack of previous literature on the subject using the same measures, sample calculation was based on a population estimate. The most extreme case was considered, and a ratio of 0.5 was assumed. 2.3. Questionnaire (see Appendix A Table A1).

A questionnaire was designed by the research team to assess motivations for COVID-19 vaccination. The questionnaire was based on a previous one conducted by Alpiñáriz et al., to study the acceptability of other vaccines, such as influenza A (H1N1) [14]. Besides it being completely anonymous and voluntary, there were no open-ended questions in the questionnaire. Sociodemographic variables included were: age, categorised according to age groups (18–28 years; 29–38 years; 39–48 years; 49–60 years); sex (male, female, non-binary); marital status (married, divorced, single, and other); level of education (primary, secondary; high school/vocational training; university; and no response), professional category (17 categories classified according to the National Institute of Statistics (INE) [15], which were subsequently grouped according to the economic sector to which they belonged, those being: the primary sector: 10 and 16; the secondary sector: 11, 12, 13, and 14; the tertiary sector: 2, 3, 4, 5, 6, 7, 8, 9, 15, and 17; and the quaternary sector: 1. The last designation included professions related to military activities). In regard to places of residence, these were classified according to whether or not they corresponded to the vaccination point located in Manresa, the main motivation for being vaccinated, and the respondent’s susceptibility to, or preoccupation with, having any post-vaccination side effects.

The administrative personnel at the vaccination sites handed out the questionnaires to attending users. If individuals agreed to answer, it was considered that they were giving their consent to participate in the study. Completion rate was 91.6%. Participants responded to the questionnaire during the recommended post-vaccine waiting time for the assessment of possible, immediate, adverse effects.

All users who had difficulty reading and understanding the questionnaire were excluded.

### 2.3. Ethical Considerations

Respondents were informed that data would be collected and analysed anonymously and results would be published in aggregate form. The study protocol was approved by the local IDIAP Jordi Gol ethics committee (Code 21/172-PCV).

### 2.4. Statistical Analysis

The questionnaires were read automatically with OpenText TeleForm v16.5 software. Then, a descriptive statistical analysis was performed for the data derived from the questionnaire responses. Categorical variables were described by frequencies and percentages. For the continuous variables, we used the mean and standard deviation. The proportions of categorical variables were compared using Fisher’s exact test and the *t*-test in the case of continuous variables.

Next, to estimate the association of the main sociodemographic variables with the reasons for vaccination, a logistic regression was applied in each case. Finally, the results of the models were presented with odds ratios (OR), *p* values, and 95% CI.

The statistical analysis was performed with the R program, version 4.0.0 (R Project for Statistical Computing; Vienna, Austria), with bilateral tests, taking the *p*-value < 0.05 as statistically significant, and using 95% confidence intervals.

## 3. Results

A total of 1485 eligible questionnaires were collected, of which 124 were rejected for the lack of a specification of the reason for vaccination, leaving a final sample of 1361 questionnaires—a 91.6% completion rate.

The mean age of the respondents was 31 years (SD 10.3), with 52.91% female. Of the respondents, 11.86% reported that they had previously been diagnosed with COVID-19. A total of 60.31% were single users, and 72.10% belonged to the tertiary production sector (Table 1).

The most common motivations for vaccination were fear of infecting family (49.52%) and fear of self-infection (39.45%), followed by socialising and travel (31.00% and 30.56%, respectively). The least common reasons for motivation for vaccination were work (21.97%) and social and/or family pressure (11.75%) (Table 2).

On the other hand, 31% of respondents indicated that they believed they had side effects, with a median of 6 (3–8) out of 10 concerns for these effects (Table 2). A logistic regression model was then applied to each reason for vaccination to estimate the associations between age, sex, marital status, educational level, production sector, mass vaccination point, previous COVID-19 infection, and COVID-19 infection of a family member.

Table 3 shows a significant inverse relationship between age and motivation to be vaccinated for travel: the older the age, the lower the odds. In the younger age group, being between 29 and 38 years old reduces the OR by 37%, between 39 and 48 years old by 53%, and between 49 and 60 years old by 69%. It was also seen that older users selected the reason of “fear of infecting family” less frequently than younger users.

Women were more motivated to be vaccinated for fear of becoming infected (*p =* 0.001; OR 1.63, IC 95% 1.22–2.18) and infecting the family (*p =* 0; OR 2.2, IC 95% 1.64–2.94) than men. Single and other marital statuses selected more motivation to be vaccinated for travel generally than married (*p =* 0.033; OR 1.59, IC 95% 1.04 to 2.45); (*p =* 0.006; OR 2.16, IC 95% 1.24 to 3.75), respectively. Regarding the level of education, both users with high school and vocational training studies and those with university studies were vaccinated more for fear of infecting family (*p =* 0.013; OR 2.82, IC 95% 1.28–6.72), (*p =* 0.019; OR 2.69, IC 95% 1.21–6.44) than users with primary school studies. University-educated users also did so for travel in general (*p =* 0.018; OR 3.77, IC 95% 1.39–13.23).

Workers in the secondary sector expressed more fear of contagion (OR 2.81; *p =* 0.01, IC 95%) than workers in the primary sector.

Users with a Spanish nationality (90.9%) were vaccinated for fear of infecting family (*p =* 0; OR 3.21, IC 95%) and for socialising (*p =* 0.029; OR 1.91, IC 95%). In addition, users who were not assigned to the Manresa population vaccination point went to be vaccinated for work reasons (*p =* 0.017; OR 0.68, IC 95%); for travelling in general (*p =* 0.021; OR 0.7, IC 95%); and also indicated less concern about side effects.

Finally, users who had not been infected with COVID-19 were vaccinated for fear of contagion (*p =* 0.022; OR 1.73, IC 95%) and fear of infecting their family (*p =* 0.024; OR 1.68, IC 95%), while those who had already had the disease were more motivated by socialising.

## 4. Discussion

The research question for the study was based on the reasons why people wanted to be vaccinated against COVID-19 in an area of Catalonia (Bages). The most frequently reported motivations were the fear of infecting family members, traveling, and socialising.

The period in which the surveys were administered at the mass vaccination point in Bages coincided with the expansion of the Delta variant (B.1.351) in Catalonia [16]. The results may suggest that the collapse of vaccination points in the more urban areas of Catalonia made it easier for citizens to access other vaccination points in Catalonia. Bages was a central point in the territory of Catalonia, and therefore, there were citizens who went to the vaccination point in this region to get vaccinated.

Some studies have shown that receiving information about vaccination from formal or informal sources may have different relationships with the decision to vaccinate or not [17]. In addition, according to public health experts, a convincing factor in favour of vaccination motivation is the wealth of activities in which users will be engage after they are vaccinated. This strategy is reflected in the data found in our study, which show fear of infecting family, contagion, travel, and socialising as the main reasons for vaccination.

On the one hand, it should be noted that, when the participants answered the questionnaire, the first cases of the Delta variant were being detected in Catalonia. In this context, it was known that the first waves of COVID-19 affected more elderly users. Among the first cases of the Delta variant, the potential lethality of this variant was not yet known, and therefore, one of the main motivations among the population to get vaccinated was probably the fear of infecting family members. Another motivation for vaccination was travel and socialising. The survey was conducted in summer, coinciding with a vacation period. There were many countries in the European Union that allowed entry with a vaccination certificate, enabling avoidance of the TAR and PCR tests. The implementation of measures such as vaccination to stop COVID-19 generated a certain sense of security among the population. Therefore, one way to be able to socialise and travel “safely” can be attributed to an increased motivation to be vaccinated [18,19]. As other studies have shown, the vaccination certificate should be accompanied by effective education and information in order to avoid promoting this false sense of security [20]. The results of our study coincide with other studies that also point to the fear of infecting family, especially parents, and the fear of infecting oneself as the main motivations for vaccination [21], followed by contributing to the greater relaxation of restrictions, allowing social contact, and having more opportunities to travel [22].

Older users who had not been infected at that time stated that their main motivation was also fear of becoming infected, which could be due to the epidemiological risk of becoming ill with COVID-19 through close contact with other people.

In terms of gender, the results show that the members of the female gender were vaccinated more often than those of the male gender. These results could be attributed to social roles and increased concern in contracting COVID-19 and its side effects [23]. However, some meta-analyses are suggest a lower motivation to vaccinate in women compared to men [24], as men are at higher risk for more severe infection [25], and women are at higher risk for persistent COVID [26].

Of those surveyed, the professional sector that was most vaccinated was the tertiary sector; this could be attributed to the fact that they work in teams and need to protect those close to them from infection, both in the workplace and in the family unit.

### 4.1. Strengths and Limitations

The most notable strength of this study is that the authors considered this questionnaire as a good tool to understand the people’s perceptions about vaccination. However, the questionnaire was conducted in a very dynamic and changing context. For this reason, it is probably the best option to conduct the questionnaire when it is necessary to establish strategies.

The study has some limitations. First, the survey was administered at a single mass vaccination point and although we could see that users from all over Catalonia attended, the findings of the study may not be externally generalised. Second, as the survey was completely voluntary, it is possible that there is a bias in the sample since those who answered were there to be vaccinated, and those who did not attend were not taken into account. On the other hand, the survey that was administered was not validated, and therefore, that may detract from its validity. However, the large volume of responses to the survey makes it a good indicator for people’s motivations regarding vaccination against COVID-19, an approach recommended by experts [26].

### 4.2. Implications

For practical purposes, given the importance of how subnational governments are responding to COVID-19 in large countries, the results of this questionnaire can be taken into account to establish strategies in specific areas to promote vaccination. In addition, for future research, this type of questionnaire could be applied in different contexts to achieve vaccination coverage for specific pathologies.

## 5. Conclusions

Motivational experience with other vaccination programmes, such as those for influenza, may serve as a strategy for improving motivation for the COVID-19 vaccine. The results show that measuring the vaccination intentions of a specific territory can be a good indicator for the vaccination coverage of the population and can be important when designing and establishing strategies aimed at specific target groups; our study can motivate studies with a more qualitative approach.

## Figures and Tables

**Table 1 vaccines-10-00597-t001:** Descriptive analysis of the sample.

Variables	*n* (%) *n* = 1361
Age	
18–28	672 (49.37)
29–38	381 (27.99)
39–48	209 (15.35)
49–60	99 (7.27)
Sex	
Male	635 (46.72)
Female	719 (52.91)
Non-binary	5 (0.37)
Marital Status	
Married	345 (25.78)
Single	807 (60.31)
Divorced	45 (3.36)
Other	141 (10.54)
Level of education	
Primary	60 (4.44)
Secondary	198 (14.64)
Baccalaureate, Vocational Training	518 (38.31)
University students	570 (42.16)
No response	6 (0.44)
Production sector	
Primary	55 (5.00)
Secondary	189 (17.18)
Tertiary	793 (72.10)
Quaternary	63 (5.73)
Nationality	
Non-Spanish	120 (9.1)
Spanish	1211 (90.9)
Mass vaccination point	
Other counties	422 (33.12)
Bages-Moianès	852 (66.87)
COVID infection	
Yes	159 (11.86)
No	1182 (88.14)
Familial COVID infection	
Yes	634 (47.78)
No	693 (52.22)

**Table 2 vaccines-10-00597-t002:** Descriptive analysis of the reasons for vaccination and side associations, for the total sample and according to the sociodemographic variables studied. The table shows the absolute frequencies and, in parentheses, the percentages. The test used was the X^2^.

	Fear of Infecting Family	Fear of Contagion	Socialising	Travel in General	Occupational	Social and Family Pressure	Concerns about Side Effects	Do You Think You Will Have It?
Total sample	674 (49.52%)	537 (39.45%)	422 (31.00%)	416 (30.56%)	299 (21.97%)	160 (11.75%)	6 [3; 8]	412 (31.00%)
IC 95%	(46.83; 52.21)	(36.86; 42.11)	(28.57; 33.55)	(28.14; 33.10)	(19.81; 24.28)	(10.12; 13.61)		(28.53; 33.58)
Sex								
Male	259 (40.8%)	213 (33.5%)	200 (31.5%)	190 (29.9%)	149 (23.5%)	82 (12.9%)	5 [2; 7]	164 (26.2%)
Female	412 (57.3%)	322 (44.8%)	219 (30.5%)	223 (31.0%)	149 (20.7%)	76 (10.6%)	7 [5; 8]	245 (35.2%)
Non-binary	2 (40.0%)	1 (20.0%)	2 (40.0%)	1 (20.0%)	0 (0.00%)	1 (20.0%)	5 [3; 19]	2 (40.0%)
*p*-value	<0.001	<0.001	0.729	0.886	0.276	0.238	<0.001	0.001
Age								
18–28	361 (53.7%)	266 (39.6%)	230 (34.2%)	261 (38.8%)	141 (21.0%)	84 (12.5%)	5 [3; 7]	218 (33.1%)
29–38	188 (49.3%)	153 (40.2%)	105 (27.6%)	97 (25.5%)	84 (22.0%)	50 (13.1%)	6 [4; 8]	115 (31.1%)
39–48	100 (47.8%)	90 (43.1%)	51 (24.4%)	43 (20.6%)	53 (25.4%)	19 (9.09%)	7 [5; 9]	59 (28.9%)
49–60	25 (25.3%)	28 (28.3%)	36 (36.4%)	15 (15.2%)	21 (21.2%)	7 (7.07%)	7 [5; 8]	20 (20.8%)
*p*-value	<0.001	0.094	0.012	<0.001	0.611	0.206	<0.001	0.094
Marital Status								
Married	158 (45.8%)	147 (42.6%)	93 (27.0%)	69 (20.0%)	79 (22.9%)	37 (10.7%)	7 [5; 9]	99 (29.8%)
Single	423 (52.4%)	314 (38.9%)	260 (32.2%)	284 (35.2%)	172 (21.3%)	94 (11.6%)	5 [3; 7]	238 (29.9%)
Divorced	15 (33.3%)	13 (28.9%)	12 (26.7%)	7 (15.6%)	5 (11.1%)	2 (4.44%)	7 [5; 8]	11 (24.4%)
Other	64 (45.4%)	48 (34.0%)	50 (35.5%)	51 (36.2%)	38 (27.0%)	22 (15.6%)	6 [4; 8]	56 (41.2%)
*p*-value	0.016	0.152	0.179	<0.001	0.138	0.194	<0.001	0.043
Level of education								
Primary	20 (33.3%)	23 (38.3%)	11 (18.3%)	5 (8.33%)	10 (16.7%)	8 (13.3%)	6 [5; 9.5]	15 (26.3%)
Secondary	86 (43.4%)	72 (36.4%)	54 (27.3%)	35 (17.7%)	41 (20.7%)	27 (13.6%)	6 [5; 8]	62 (32.1%)
Baccalaureate, Vocational Training	272 (52.5%)	202 (39.0%)	161 (31.1%)	167 (32.2%)	129 (24.9%)	64 (12.4%)	6 [3.75; 8]	159 (31.4%)
University students	291 (51.1%)	235 (41.2%)	194 (34.0%)	206 (36.1%)	116 (20.4%)	59 (10.4%)	6 [3; 8]	171 (30.6%)
Dk/No	1 (16.7%)	0 (0.00%)	0 (0.00%)	1 (16.7%)	0 (0.00%)	1 (16.7%)	9 [7.25; 10]	3 (60.0%)
*p*-value	0.006	0.244	0.026	<0.001	0.217	0.552	0.011	0.611
Production sector								
Primary	25 (45.5%)	13 (23.6%)	17 (30.9%)	15 (27.3%)	14 (25.5%)	10 (18.2%)	6 [3; 8]	15 (28.3%)
Secondary	89 (47.1%)	82 (43.4%)	62 (32.8%)	54 (28.6%)	49 (25.9%)	24 (12.7%)	5 [3; 8]	50 (27.2%)
Tertiary	402 (50.7%)	304 (38.3%)	225 (28.4%)	245 (30.9%)	189 (23.8%)	92 (11.6%)	6 [4; 8]	262 (33.6%)
Quaternary	26 (41.3%)	21 (33.3%)	17 (27.0%)	17 (27.0%)	20 (31.7%)	5 (7.94%)	6 [3; 8]	18 (29.5%)
*p*-value	0.402	0.052	0.642	0.814	0.538	0.363	0.085	0.330
Nationality								
Non-Spanish	36 (30.0%)	47 (39.2%)	20 (16.7%)	42 (35.0%)	30 (25.0%)	12 (10.0%)	6 [4.5; 8]	31 (26.3%)
Spanish	627 (51.8%)	479 (39.6%)	394 (32.5%)	366 (30.2%)	264 (21.8%)	143 (11.8%)	6 [3; 8]	370 (31.3%)
*p*-value	<0.001	1	0.001	0.328	0.490	0.660	0.154	0.306
Mass vaccination point								
Other counties	230 (54.5%)	180 (42.7%)	141 (33.4%)	157 (37.2%)	107 (25.4%)	56 (13.3%)	5 [3; 8]	124 (30.1%)
Bages-Moianès	408 (47.9%)	328 (38.5%)	264 (31.0%)	239 (28.1%)	175 (20.5%)	93 (10.9%)	6 [3; 8]	261 (31.3%)
*p*-value	0.031	0.172	0.417	0.001	0.061	0.255	0.043	0.715
COVID infection								
Yes	56 (35.2%)	49 (30.8%)	63 (39.6%)	47 (29.6%)	38 (23.9%)	14 (8.81%)	6 [3.5; 8]	42 (27.1%)
No	615 (52.0%)	485 (41.0%)	357 (30.2%)	368 (31.1%)	255 (21.6%)	141 (11.9%)	6 [3; 8]	366 (31.7%)
*p*-value	<0.001	0.017	0.021	0.755	0.573	0.306	0.407	0.286
Familial COVID infection								
Yes	308 (48.6%)	255 (40.2%)	211 (33.3%)	208 (32.8%)	143 (22.6%)	77 (12.1%)	6 [3; 8]	205 (33.4%)
No	354 (51.1%)	271 (39.1%)	206 (29.7%)	201 (29.0%)	147 (21.2%)	76 (11.0%)	6 [3; 8]	199 (29.1%)
*p*-value	0.392	0.720	0.182	0.150	0.600	0.558	0.524	0.103

The variable “concern for side effects” has been described by the median, and the first and third quartiles in “square brackets”, and the comparison has been made using the Mann–Whitney test.

**Table 3 vaccines-10-00597-t003:** Logistic regression.

	Occupational			Travel in General			Fear of Contagion			Fear of Infecting Family		
Variables	GOLD	IC 95%	*p*-Value	GOLD	IC 95%	*p*-Value	GOLD	IC 95%	*p*-Value	GOLD	IC 95%	*p*-Value
Age												
29–38	0.93	(0.63; 1.36)	0.695	0.63	(0.44; 0.89)	0.011	0.97	(0.69; 1.37)	0.871	0.82	(0.59; 1.16)	0.268
39–48	1.46	(0.89; 2.4)	0.131	0.47	(0.28; 0.78)	0.004	1.23	(0.78; 1.94)	0.377	0.72	(0.46; 1.14)	0.167
49–60	1.23	(0.61; 2.41)	0.554	0.31	(0.14; 0.65)	0.003	0.56	(0.29; 1.07)	0.084	0.34	(0.17; 0.64)	0.001
Female	0.88	(0.64; 1.22)	0.436	1.04	(0.76; 1.41)	0.816	1.63	(1.22; 2.18)	0.001	2.2	(1.64; 2.94)	<0.001
Marital Status												
Single	1.17	(0.77; 1.78)	0.466	1.59	(1.04; 2.45)	0.033	0.71	(0.49; 1.03)	0.069	0.88	(0.61; 1.28)	0.512
Divorced	0.35	(0.1; 0.95)	0.061	1.16	(0.41; 2.89)	0.758	0.62	(0.28; 1.34)	0.235	0.83	(0.38; 1.79)	0.644
Other	1.56	(0.9; 2.67)	0.107	2.16	(1.24; 3.75)	0.006	0.72	(0.43; 1.2)	0.208	0.79	(0.47; 1.32)	0.374
Level of education												
Secondary	1.01	(0.43; 2.62)	0.975	1.44	(0.5; 5.28)	0.532	0.93	(0.42; 2.12)	0.861	2.02	(0.88; 4.98)	0.108
Baccalaureate, Vocational Training	1.31	(0.59; 3.24)	0.528	2.7	(1; 9.46)	0.075	1.04	(0.49; 2.27)	0.925	2.82	(1.28; 6.72)	0.013
University students	0.89	(0.39; 2.23)	0.796	3.77	(1.39; 13.23)	0.018	1.2	(0.57; 2.66)	0.634	2.69	(1.21; 6.44)	0.019
Production sector												
Secondary	1.27	(0.6; 2.85)	0.551	1.29	(0.6; 2.92)	0.523	2.81	(1.32; 6.45)	0.01	1.28	(0.63; 2.63)	0.487
Tertiary	1.2	(0.59; 2.62)	0.626	0.98	(0.47; 2.12)	0.952	1.92	(0.93; 4.28)	0.09	0.89	(0.46; 1.75)	0.739
Quaternary	2.06	(0.84; 5.25)	0.119	1.17	(0.46; 3.04)	0.744	1.69	(0.68; 4.38)	0.264	0.82	(0.35; 1.94)	0.658
Nationality: Spanish	0.83	(0.5; 1.44)	0.502	0.77	(0.46; 1.29)	0.307	0.92	(0.57; 1.49)	0.721	3.21	(1.94; 5.46)	<0.001
RRP: Bages-Moianès	0.68	(0.5; 0.93)	0.017	0.7	(0.52; 0.95)	0.021	0.83	(0.63; 1.11)	0.218	0.79	(0.59; 1.05)	0.107
COVID infection: No	0.93	(0.57; 1.53)	0.767	0.75	(0.46; 1.23)	0.246	1.73	(1.09; 2.78)	0.022	1.68	(1.07; 2.66)	0.024
Familial COVID infection: No	0.93	(0.68; 1.27)	0.641	0.78	(0.58; 1.05)	0.100	0.79	(0.6; 1.04)	0.087	0.92	(0.7; 1.21)	0.539

Reference categories: 18–28 years old, male, married, primary education level, primary sector, non-Spanish nationality, from outside Bages-Moianès, with COVID infection, and with COVID infection in a family member.

## Data Availability

The data presented in this study are available upon request from the corresponding author.

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
