# Peer review of "Motivation for Vaccination against COVID-19 in Persons Aged between 18 and 60 Years at a Population-Based Vaccination Site in Manresa (Spain)"

_vaccines, 2022, doi:10.3390/vaccines10040597_

Round 1
Reviewer 1 Report
Comments on Manuscript Title “Motivation for vaccination against COVID-19 in persons aged 18 years and older at a Population-based vaccination site. Observational, descriptive, cross-sectional study”
Please below are comments for the authors of manuscript
Title:
- Please delete Observational, descriptive, cross-sectional study. The new title will be “Motivation for vaccination against COVID-19 in persons aged 18 years and older at a Population-based vaccination site in ------------ (Location and country)
- Also, I think this work assessed vaccination sites users under 60 years of age. Saying 18 years and older may be misleading. I think the title should be rephrased. You may wish to delete age from the title and describe in aim and methods section. Please, I hope I am not missing something here. “Motivation for vaccination against COVID-19 in persons aged between 18 and 60 years at a Population-based vaccination site in ------------ (location/city and country)
Abstract:
- Cross-sectional descriptive study “cross-sectional study or survey”. Please delete descriptive
- Line 16- add Spain
- Line 17 - Instead of “some” use “A total of 1,361”
- Line 17 – surveys? were analysed. I think you should use “questionnaires”.
- Line 17- analysed using what methods?
- Lines 20-22- are saying there was a relationship between each of these variables (age, sex, marital status, educational level, production sector, mass vaccination point, previous COVID-19 infection and COVID-19 infection of a family member) and motivation. Please provide atleast the p values
- Lines 22-23. Please make the relevance of the work clearer. E.g. The data generated will inform decisions and formulations of appropriate campaigns that will promote vaccination in specific population groups.
Introduction
- Line 29 remove global (repetition)
- Line 68: “to be vaccinated” seems more appropriate based on the aim/concept of the study
Materials and methods
- Line 80 – Please clarify this “older than 18 years and younger than 60 years”. The table 1 on descriptive analysis of respondents, and age group categorization showed otherwise. Did the survey involve respondents within age range 18 and 60?
- Line 83-84. Please move to questionnaire section. May be on the last paragraph
- Subsection on sample: It will be great for the authors to expatiate more on how the sample size was estimated? What software package or formula used? Why a precision of 3%?
- Please provide in detail your recruitment process. How many individuals were approached? Response rate? How were individuals selected? What were the inclusion and exclusion criteria for participants?
- Line 87: It was estimated that 15% of replacements were needed? How? For what reasons?
- Questionnaire: Was this design based on past literature or experiences? Line 90.
- The authors need to let us know more about the validity of the questionnaire? Was a pretest or validation conducted? How many questions were open or closed ended? How did the authors seek respondents consent? Verbal or written? What is the questionnaire completion rate as well? Move Line 83-84 to conclude the last paragraph of the questionnaire subsection
- Statistical analysis: Continuous variables were described with the mean and standard deviation- what informed the use of these two parameters?
- Line 117: Logistic regression is a predictive modeling technique which can be used to establish an association or relationship between dependent variables and one or several independent variables. I will suggest “the effect” be changed to association or relationship. Cross-sectional studies do not determine effects rather determine the relationships between explanatory variables and the corresponding outcome variable. Please amend throughout the manuscript
- Line 119, OR, p values and 95% CI. Include all
Results
- Line 125: “surveys” Please use appropriate term here “eligible questionnaires”
- Lines 131-136: You may not need the CI since they are already displayed in table 2
- The logistic regression please report OR, 95% CI and p values for all the variables associated with motivation to get vaccinated
Discussion
- This section has under discussed your interesting results. Please discuss why fear of infecting family, contagion, travel and socialization are reported as the main reasons for getting the vaccines. The discussion is very inadequate compared with your study outcomes
- We need to know as part of your limitations that this may not be externally generalized.
Author Response
Comments and Suggestions for Authors
Comments on Manuscript Title “Motivation for vaccination against COVID-19 in persons aged 18 years and older at a Population-based vaccination site. Observational, descriptive, cross-sectional study”
Please below are comments for the authors of manuscript
TITLE
Point 1. Please delete Observational, descriptive, cross-sectional study. The new title will be “Motivation for vaccination against COVID-19 in persons aged 18 years and older at a Population-based vaccination site in ------------ (Location and country)
Thank you, we have changed it.
Point 2. Also, I think this work assessed vaccination sites users under 60 years of age. Saying 18 years and older may be misleading. I think the title should be rephrased. You may wish to delete age from the title and describe in aim and methods section. Please, I hope I am not missing something here. “Motivation for vaccination against COVID-19 in persons aged between 18 and 60 years at a Population-based vaccination site in ------------ (location/city and country)
Thank you for your valued suggestions. We have amended the title.
ABSTRACT
Point 3. Cross-sectional descriptive study “cross-sectional study or survey”. Please delete descriptive
You are right, sorry. We have deleted “descriptive”.
Point 4. Line 16- add Spain
Thank you, we have added.
Point 5. Line 17 - Instead of “some” use “A total of 1,361”
Thank you, we have changed it.
Point 6. Line 17 – surveys? were analysed. I think you should use “questionnaires”.
Thank you, we have changed surveys for “questionnaires”.
Point 7.Line 17- analysed using what methods?
The methodology is a cross-sectional study and we used statistical analysis.
Point 8. Lines 20-22- are saying there was a relationship between each of these variables (age, sex, marital status, educational level, production sector, mass vaccination point, previous COVID-19 infection and COVID-19 infection of a family member) and motivation. Please provide at least the p values
We just mentioned that we estimated the effect of these variables, but here we didn’t say that there was a relationship. In fact, there is not a relation with most of them as showed in the results section.
Point 9. Lines 22-23. Please make the relevance of the work clearer. E.g. The data generated will inform decisions and formulations of appropriate campaigns that will promote vaccination in specific population groups.
We have now made the relevance of the work clearer.
Introduction
Point 10. Line 29 remove global (repetition)
Thank you, we have removed
Point 11. Line 68: “to be vaccinated” seems more appropriate based on the aim/concept of the study
Thank you, we have rephrased it.
MATERIALS AND METHODS
Point 12 Line 80 – Please clarify this “older than 18 years and younger than 60 years”. The table 1 on descriptive analysis of respondents, and age group categorization showed otherwise. Did the survey involve respondents within age range 18 and 60?
Yes, the age group is between 18 and 60 years. We have amended this.
Point 13. Line 83-84. Please move to questionnaire section. May be on the last paragraph
We have moved this section
Point 14. Subsection on sample: It will be great for the authors to expatiate more on how the sample size was estimated? What software package or formula used? Why a precision of 3%?
The sample size was estimated using Grammo software (version 7.12, IMIM, Barcelona, Spain) Sample calculation was based on a population estimate. The authors did a previous search and there was no literature on the subject using the same measures, so the most extreme case was considered and a ratio of 0.5 was assumed. The level of precision was 3% because it was considered that it could be a correct value for the width of the confidence interval.
We have added this information in the manuscript.
Point 15. Please provide in detail your recruitment process. How many individuals were approached? Response rate? How were individuals selected? What were the inclusion and exclusion criteria for participants?
Thank you for your comment. We obtained 1458 questionnaires, but 124 questionnaires were excluded because they were incomplete (for example the reason for vaccination was not indicated). Finally, we were able to analyze 1361 questionnaires.
When we calculated the sample, we estimated a percentage of replacements needed.
The administrative staff in the mass vaccination center invited all attendants between 18 and 60 years of age to participate in the study while they were waiting to get vaccinated.
We included all consenting patients aged 18 to 60 years and excluded those who did not understand the language or were unable to respond. Line 83
We have added this in the manuscript.
Point 16. Line 87: It was estimated that 15% of replacements were needed? How? For what reasons?
Checking we have seen that the replacement % was 20% and not 15%. Apart from this, the calculation of the sample was estimated with this % of replacement because, being a questionnaire filled in by the users themselves, it is to be expected that a % of questionnaires are not well answered, that relevant information is missing and therefore cannot be analyzed, or that the program with which the reading the questionnaires does not understand some of the answers. Therefore, it was decided to overestimate the total sample in case some of the user surveys should need to be removed afterwards.
The calculation was made with the calculator as stated in the previous question
Point 17. Questionnaire: Was this design based on past literature or experiences? Line 90.
This is a very valuable comment, thank you. The questionnaire was based on a previous one conducted by Apiñániz et al (14) to study the acceptability of influenza A (H1N1) vaccine. This original questionnaire was not informed by any theories. The authors considered it was suitable for comparing the perceptions of citizens for the coronavirus vaccination and adapted it.
We have added this information and a new reference:
14- Apiñániz A, López-Picado A, Miranda-Serrano E, et al. Population-based cross-sectional study on vaccine acceptability and perception of A/H1N1 influenza severity: Opinion of the general population and health professionals. Gac Sanit. 2010;24(4):314-320. doi: 10.1016/j.gaceta.2010.03.009
Point 18. The authors need to let us know more about the validity of the questionnaire? Was a pretest or validation conducted? How many questions were open or closed ended? How did the authors seek respondents consent? Verbal or written? What is the questionnaire completion rate as well? Move Line 83-84 to conclude the last paragraph of the questionnaire subsection
As mentioned before, the questionnaire is based on a previous published questionnaire. Apiñániz questionnaire was adapted from an earlier survey conducted in Hong Kong. The authors considered that the Apiñániz survey was more appropriate (because it was used in Spanish population) than the previous one. In addition, the authors changed some questions in order to respond the main objective. There were no open questions in the questionnaire.
The questionnaire was completely anonymous and voluntary.
The administrative staff in the mass vaccination center verbally invited attendants to participate in the study while they were waiting to get vaccinated. If the individual agreed to answer, we considered that they were giving their consent to participate in the study.
Completion rate was 91,6%.
We have added this information in the manuscript.
Point 19. Statistical analysis: Continuous variables were described with the mean and standard deviation- what informed the use of these two parameters?
We only used the median and deviation by age. Other numerical variables, “concerns about side effects” we used the median and the 1st and 3rd quartiles, since it was a scale and the distribution of the scale was not normal.
Point 20. Line 117: Logistic regression is a predictive modeling technique which can be used to establish an association or relationship between dependent variables and one or several independent variables. I will suggest “the effect” be changed to association or relationship. Cross-sectional studies do not determine effects rather determine the relationships between explanatory variables and the corresponding outcome variable. Please amend throughout the manuscript
Thank you, We have amended in the manuscript
Point 21. Line 119, OR, p values and 95% CI. Include all
We have included it.
RESULTS
Point 22. Line 125: “surveys” Please use appropriate term here “eligible questionnaires”
Thank you, we have used eligible questionnaires.
Point 23. Lines 131-136: You may not need the CI since they are already displayed in table 2
Thank you, we have now removed CI in this section.
Point 24. The logistic regression please report OR, 95% CI and p values for all the variables associated with motivation to get vaccinated
We have amended in the manuscript
DISCUSSION
Point 25. This section has under discussed your interesting results. Please discuss why fear of infecting family, contagion, travel and socialization are reported as the main reasons for getting the vaccines. The discussion is very inadequate compared with your study outcomes
This is a very valuable comment, thank you so much. We have added this in the discussion:
“On the one hand, it should be noted that when the participants answered the questionnaire, the first cases of the Delta variant were detected in Catalonia. In this context, it was known that the first waves of COVID-19 affected more elderly users. Among the first cases of the Delta variant, the potential lethality of this variant was not yet known and therefore, one of the main motivations among the population to get vaccinated was probably the fear of infecting family members. On the other hand, another of the main motivations for vaccination was travel and socialization. The survey was conducted in summer, coinciding with a vacation period. There were many countries in the European Union that allowed entry into the country with the vaccination certificate, avoiding the TAR and PCR tests. The implementation of measures such as vaccination to stop COVID-19 generated a certain sense of security among the population. Therefore, one way to be able to socialize and travel "safely" can be attributed to an increased motivation to get vaccinated.”
Point 26. We need to know as part of your limitations that this may not be externally generalized.
Thank you, indeed this is a limitation. We have now added this in the appropriate section (limitations).

Reviewer 2 Report
General
Authors explore in their manuscript ‘Motivation for vaccination against COVID-19 in persons aged 18 years
and older at a population-based vaccination site. Observational, descriptive, cross-sectional study’ the motivation for vaccination against Covid-19. Such studies are not done that often, however some work should be done before this manuscript can be published.
Title
Please change the title to a shorter one.
Authors
Glòria Sauch Valmaña1,2,* Aïna Fuster-Casanovas 1 Anna Ramírez-Morros 1 Berta Rodoreda Pallàs 1,2 Josep Vidal- Alaball 1,2,3 Anna Ruiz-Comellas 1,2,3 i Queralt Miró Catalina 1,2
- Add a comma between the names
- Replace <i> by ‘and’
Abstract
Background
Why do you want to know this?
(please also compare the sentence in the Abstract (below 60) and the title (over 18); it does not fit)
Methods
Please make the readership clear whether you used only people who were vaccinated.
Results
Explain the readership what you mean with <mass vaccination point>, as only one such point is used?
Conclusion
-
Key words
-
Introduction
-
Methods
Sample
I don't understand your phrases: <A sample of 1,333 individuals was estimated to be necessary to estimate the reasons for vaccination with a precision of 3 percentage points and 95% confidence intervals. It was estimated that 15% of replacements were needed.>
Measures
I have a question to your sentence <An ad hoc questionnaire>. Is your Statistics Spain not having ready made questions about age, sex, marital status, level of education? Then it is not necessary to categorize these elements as ‘ad hoc’.
Please change <dk/no> into something understandable.
Statistical analyses
Please rewrite this section: First we … . Then we … . Next we … . Finally we … . The readership easier grabs what you did and in which order the Results will be shown.
Results
Please change <11.86% had had COVID-19.> into something indicating that they reported this themselves or that this message was objectivated e.g. ‘11.86% reported that they had had COVID-19.’
Please add the p-value to the next sentence <Women were more motivated to be vaccinated for fear of becoming infected (p0.001; OR 1.63) and infecting the family (p0;OR 2.2) than men.>
Discussion
Please keep in mind the following structure for writing a Discussion:
para1 start with repeating the research question + answer this (you now know the outcomes) without any comments or interpretation.
Para2,3,# start a new para, 1 topic per para, and start this para with one of your findings – which then defines the content of the para. Relate your finding to earlier published references.
Strengths and limitations
Mention strengths! Limitations are those issues which might bias your findings (give the direction)(are there sources of bias, like information bias or selection bias and how did you handle confounders?).
Implications
(split into: for practice, for future research)
Conclusion
Please rewrite this Discussion part
Tables, Figures
Last table: please translate in English!

Author Response
General
Authors explore in their manuscript ‘Motivation for vaccination against COVID-19 in persons aged 18 years
and older at a population-based vaccination site. Observational, descriptive, cross-sectional study’ the motivation for vaccination against Covid-19. Such studies are not done that often, however some work should be done before this manuscript can be published.
Title
Please change the title to a shorter one.
We have amended and shortened the title taking also in consideration another reviewer’s comments
Authors
Glòria Sauch Valmaña1,2,* Aïna Fuster-Casanovas 1 Anna Ramírez-Morros 1 Berta Rodoreda Pallàs 1,2 Josep Vidal- Alaball 1,2,3 Anna Ruiz-Comellas 1,2,3 i Queralt Miró Catalina 1,2
- Add a comma between the names
- Replace <i> by ‘and’
We have now amended this
Abstract
Background
Why do you want to know this?
(please also compare the sentence in the Abstract (below 60) and the title (over 18); it does not fit)
We have now amended this
Methods
Please make the readership clear whether you used only people who were vaccinated.
This is a provisional space outside the health centre that was set up in Catalonia in the context of the Covid-19 pandemic, where people from all over Catalonia could go to get vaccinated for Covid-19 by appointment.
We have explained in the manuscript.
Results
Explain the readership what you mean with <mass vaccination point>, as only one such point is used?
This is a good point, we have included the description in the manuscript.
Conclusion
-Key words
-Introduction
Methods
Sample
I don't understand your phrases: <A sample of 1,333 individuals was estimated to be necessary to estimate the reasons for vaccination with a precision of 3 percentage points and 95% confidence intervals. It was estimated that 15% of replacements were needed.>
This is a good point, thank you for your suggestion. The calculation was based on a population estimate and as we did not enter in the bibliography studies in which the same measures were used. We considered the most extreme case, assuming a proportion of 0.5.
The level of precision was 3% because we considered this was a correct value for the with of the confidence interval.
We have realised we made a mistake. Replacements were around 20% and not 15% as we indicated. We made an estimate of the sample with this replacement for the characteristics of the questionnaires as many could be incomplete or the reading system would not understand the answer. For all these reasons we decided to overestimate the sample in case some questionnaires had to be eliminated
Measures
I have a question to your sentence <An ad hoc questionnaire>. Is your Statistics Spain not having ready made questions about age, sex, marital status, level of education? Then it is not necessary to categorize these elements as ‘ad hoc’.
Please change <dk/no> into something understandable.
This is a good point, thank you. Perhaps we have not use this term correctly; the questionnaire was based on a previous one (past literature) conducted by Apiñániz et al (14) to study the acceptability of influenza A (H1N1) vaccine. This original questionnaire was not informed by any theories. The authors considered it was suitable for comparing the perceptions of citizens for the coronavirus vaccination and adapted it.
14- Apiñániz A, López-Picado A, Miranda-Serrano E, et al. Population-based cross-sectional study on vaccine acceptability and perception of A/H1N1 influenza severity: Opinion of the general population and health professionals. Gac Sanit. 2010;24(4):314-320. doi: 10.1016/j.gaceta.2010.03.009
Statistical analyses
Please rewrite this section: First we … . Then we … . Next we … . Finally we … . The readership easier grabs what you did and in which order the Results will be shown.
Thank you for your comment. We have re-wrote this section.
Results
Please change <11.86% had had COVID-19.> into something indicating that they reported this themselves or that this message was objectivated e.g. ‘11.86% reported that they had had COVID-19.’
Ok, we have changed it.
Please add the p-value to the next sentence <Women were more motivated to be vaccinated for fear of becoming infected (p0.001; OR 1.63) and infecting the family (p0;OR 2.2) than men.>
Thank you, we have added
Discussion
Please keep in mind the following structure for writing a Discussion:
para1 start with repeating the research question + answer this (you now know the outcomes) without any comments or interpretation.
Thank you, we have done it.
Para2,3,# start a new para, 1 topic per para, and start this para with one of your findings – which then defines the content of the para. Relate your finding to earlier published references.
Thank you, we have amended:
1st para: the research question + answer
2nd para: Why people attended to this region of Catalonia to get the vaccine
3rd para: information sources and decision to get vaccinated.
4rd para: discussion about the most commented motivation
5thpara: considerations in elderly people
6th para: considerations in terms of gender
7th para: considerations in professional sector
Strengths and limitations
Mention strengths! Limitations are those issues which might bias your findings (give the direction)(are there sources of bias, like information bias or selection bias and how did you handle confounders?).
Thank you, we have added
Implications
(split into: for practice, for future research)
Thank you, we have added the implications.
Conclusion
Thanks for your response we have included and amended your suggestions in this manuscript
Please rewrite this Discussion part
Tables, Figures
Last table: please translate in English!
Thank you, we have translated the last table in English.

Reviewer 3 Report
The aim of this study was to examine patients’ motivation to vaccinate against COVID-19 at a single mass vaccination point in Catalonia. The study included 1361 adult subjects under the age of 60 years who fulfilled the questionnaire The most common reasons for vaccination included concerns of infecting the family or self-infection, socialization and travel. The motivation differed according to age, sex, education level and occupation. The results are discussed in the context of results obtained by others.
The topic and the results are of interest and the manuscript is well-written. I have no specific critical comments.
Author Response
Dear Review,
Thank you for your revision
Regards

Reviewer 4 Report
First of all, I would like to thank for the opportunity to review this paper. The ongoing pandemic has resulted in global health, economic and social crises. Actually, the vaccination campaign is the first method to counteract the COVID-19 pandemic; however, sufficient vaccination coverage is conditioned by the people’s acceptance of these vaccines in the general population. In this context, the paper under review is aimed at identifing main motivations to get vaccinated against COVID-19 in a population of over 18 years old in Spain in 2021.
The subject under study is certainly important, especially in the historical period we are experiencing. The article presents interesting results but the manuscript must be improved especially for the local impact and different epidemiological context in which it was carried on. I would like to encourage authors to consider several issues to be improved.
Title: it must me improved, highlighting the main objective of the study and where it was performed.
Introduction: The authors should make clearer what is the gap in the literature that is filled with this study. The authors do not frame their study within the vast body of literature that addressed the issue of vaccine acceptance and vaccine hesitancy during the pandemic in young and old adults and why they choose to study over-18years old people. What is the international situation regarding the acceptance of the vaccination in the adult population?
Methods: It is not clear where the study was performed, in a city, a province a region? All the Spain? Please describe the place with the relative demographic characteristic. Sample: the Author report that that 1333 individuals were estimated to be necessary, but what is the reference population? And how large was it? The enrolment procedure must be better specified. How was the sample enrolled? How did the authors choose the way to enroll the sample? How did they avoid the selection bias? The survey was conducted using a non-standard questionnaire. The use of an unreliable instrument is a serious and irreversible limitation of the study. Moreover, no mention to a validation process is reported. What about face validity, reliability and intelligibility?
Statistical analysis: I suggest to insert a measure of the magnitude of the effect for the comparisons. Please consider to include effect sizes.
Discussion: I also suggest expanding, emphasizing what is the possible international contribution of the study to the literature. The discussion must be updated including the debated argument of a green pass linked to vaccination practice; if this issue was not considered by the author in the questionnaire, a paragraph should be added in the limit section with a proper reference
Author Response
Title: it must be improved, highlighting the main objective of the study and where it was performed.
We have amended the title also taking also in consideration another reviewer’s comments
Introduction:
The authors should make clearer what is the gap in the literature that is filled with this study. The authors do not frame their study within the vast body of literature that addressed the issue of vaccine acceptance and vaccine hesitancy during the pandemic in young and old adults and why they choose to study over-18years old people. What is the international situation regarding the acceptance of the vaccination in the adult population?
It is not that there is a gap in the literature that this study aims to fill. The aim of this study is to highlight the tools we have around us to be able to establish policies aligned with increasing vaccination coverage in a determined region. As you rightly comment, the authors do not frame this study in the acceptance or non-acceptance of vaccines. The aim of this study is to study the reasons why the population accepts the vaccine. It was decided to carry out the questionnaire on people over 18 years of age, since parental consent was required for minors.
Methods: 
It is not clear where the study was performed, in a city, a province a region?
The study was performed in Manresa. It’s the capital of the area of Bages in Catalonia, Spain.
All the Spain?
No, only one of forty-two points that it was available in Catalonia, this subject we have included in the manuscript to better understanding.
Please describe the place with the relative demographic characteristic. Sample: the Author report that that 1333 individuals were estimated to be necessary, but what is the reference population?
The questionnaire was conducted in an area of Catalonia, Bages. It has got 175.527 inhabitants and it has composited by thirty countries; however, all the citizens in Catalonia could go in this area to get the vaccine.
And how large was it?
Bages area has 1092 km2 of the extension
The enrolment procedure must be better specified. How was the sample enrolled?
How did the authors choose the way to enroll the sample? How did they avoid the selection bias?
People who wanted to get the vaccine were invited by the professionals. Thus, professionals gave the questionnaire to the people and they could answer. It has to take into account that in the mass vaccination point could go all citizens who wanted to get the vaccine in Catalonia. For this reason, the authors had considered that we could have a representative sample of the population and we could minimize the selection bias.  
The survey was conducted using a non-standard questionnaire. The use of an unreliable instrument is a serious and irreversible limitation of the study. Moreover, no mention to a validation process is reported. What about face validity, reliability and intelligibility?
This is a good point, thank you. The questionnaire was based on a previous one conducted by Apiñániz et al (14) to study the acceptability of influenza A (H1N1) vaccine. This original questionnaire was not informed by any theories. The authors considered it was suitable for comparing the perceptions of citizens for the coronavirus vaccination and adapted it.
We have added this information and a new reference:
14- Apiñániz A, López-Picado A, Miranda-Serrano E, et al. Population-based cross-sectional study on vaccine acceptability and perception of A/H1N1 influenza severity: Opinion of the general population and health professionals. Gac Sanit. 2010;24(4):314-320. doi: 10.1016/j.gaceta.2010.03.009
Statistical analysis: I suggest to insert a measure of the magnitude of the effect for the comparisons. Please consider to include effect sizes.
In the table 3, we have included OR as measure of the magnitude, anyway we do not quite understand what it refers to if it is not the OR
Discussion: I also suggest expanding, emphasizing what is the possible international contribution of the study to the literature. The discussion must be updated including the debated argument of a green pass linked to vaccination practice; if this issue was not considered by the author in the questionnaire, a paragraph should be added in the limit section with a proper reference
Thank you for your value comment. We have extended the discussion.

Round 2
Reviewer 1 Report
Thank you very much for your response to the comments and suggestions made to your manuscript
Line 205- Please remove "on the other hand". The sentence will now read "Another motivation for vaccination was travel and socialization".
Author Response
Thank you very much for your response to the comments and suggestions made to your manuscript
Line 205- Please remove "on the other hand". The sentence will now read "Another motivation for vaccination was travel and socialization".
Thank you so much. We have removed in the manuscript.

Reviewer 4 Report
The paper was improved but I suggest to improve also discussion with updated references of studies conduced in the same period (see DOI: https://doi.org/10.3390/vaccines9111222)
Author Response
The paper was improved but I suggest to improve also discussion with updated references of studies conduced in the same period (see DOI: https://eur03.safelinks.protection.outlook.com/?url=https%3A%2F%2Fdoi.org%2F10.3390%2Fvaccines9111222&data=04%7C01%7Cgsauch.cc.ics%40gencat.cat%7C61f724f209934d443d9f08da1928fe39%7C3b9427dcd30e43bc8c06ff7253676fec%7C1%7C0%7C637849961112012888%7CUnknown%7CTWFpbGZsb3d8eyJWIjoiMC4wLjAwMDAiLCJQIjoiV2luMzIiLCJBTiI6Ik1haWwiLCJXVCI6Mn0%3D%7C3000&sdata=Nm%2FeZvp5rakYdKKNSNV%2BL0F91mWOQpqYOKcu8dDRS3A%3D&reserved=0
Thank you so much, it is a very valuable comment. We have added this reference and we have expanded our discussion:
“As other studies have shown, the vaccination certificate should be accompanied by effective education and information in order to avoid this false sense of security [20]”

Round 3
Reviewer 4 Report
The paper was really improved and it is suitable for pubblication